# Risk Factors for Non-Adherence to Inhaled Corticosteroids in Preschool Children with Asthma

**DOI:** 10.3390/children10010043

**Published:** 2022-12-25

**Authors:** Louise Mandrup Bach, Sune Rubak, Adam Holm-Weber, Julie Prahl, Mette Hermansen, Kirsten Skamstrup Hansen, Bo Chawes

**Affiliations:** 1Department of Pediatric and Adolescent Medicine, Herlev and Gentofte Hospital, University of Copenhagen, 2730 Herlev, Denmark; 2Danish Center of Pediatric Pulmonology and Allergology, Department of Pediatrics and Adolescents Medicine, Institute of Clinical Medicine, University Hospital of Aarhus, University of Aarhus, 8000 Aarhus, Denmark; 3Allergy Clinic, Herlev and Gentofte Hospital, University of Copenhagen, 2820 Gentofte, Denmark

**Keywords:** adherence, asthma, preschool children, risk factors, inhaled corticosteroids

## Abstract

Non-adherence to asthma controllers increases morbidity among school-aged children. This study aimed to determine non-adherence risk factors in preschool children with asthma. We investigated 172 children <6 years diagnosed with asthma in 2018 and analyzed baseline characteristics and loss of control events extracted from the medical records for four years following diagnosis. At end of follow-up, 79 children had a prescription of inhaled corticosteroids (ICS) and were included in the analyses. Adherence was assessed in a two-year period through pharmacy claims using percentage of days covered (PDC) analyzed dichotomously with non-adherence defined as PDC < 80% and using adherence ratio (AR) defined as days with medical supply divided by days without. Of the 79 children, 59 (74.7%) were classified as non-adherent. In analyses adjusted for sex, age and exacerbations prior to inclusion, adherence was positively associated with having had a loss of control event requiring a step-up in asthma controller (_a_AR:2.34 [1.10;4.98], *p* = 0.03), oral corticosteroids (_a_AR:2.45 [1.13;5.34], *p* = 0.026) or redeeming a short-acting b2-agonist prescription (_a_AR:2.91 [1.26;6.74], *p* = 0.015). Further, atopic comorbidity was associated with increased adherence (_a_AR:1.18 [1.01;1.37], *p* = 0.039), whereas having a first degree relative with asthma was associated with worse adherence (_a_AR:0.44 [0.23;0.84], *p* = 0.015). This study found poor adherence to ICS among three quarters of preschool children with asthma. Increasing adherence was associated with atopic comorbidity and loss of control events, whereas lower adherence was associated with atopic predisposition. These findings should be considered to improve adherence in preschool children with asthma.

## 1. Introduction

Asthma is the most common chronic airway disease in children [1]. Preventive treatment with inhaled corticosteroids (ICS) is the cornerstone treatment of asthma, which is administered daily and required also when the child is asymptomatic. ICS is also the primary asthma controller of choice for children of 5 years and younger although the evidence base is still rather limited and treatment decisions in some cases rely on expert opinions [2]. Maintaining adherence to ICS may depend on choice of inhaler device and is generally challenging as studies show poor adherence in approximately 50% of children with asthma aged 8–16 years [3,4,5,6].

Adherence to preventive treatment in children and adolescents decreases with increasing age, where managing the treatment relies less on the parents and more on the child, who gains more treatment responsibility [7,8]. Among preschool children, the parents are responsible for administering treatment [9] and may by prone to reduce or eliminate controller medication in the absence of symptoms, particularly due to fear of side effects of ICS. Adherence to asthma controllers, calculated as the doses administered divided by the total amount prescribed using an electronic monitor attached to the inhaler, showed a median of 70.5% among children aged 18 months to 7 years in one study [10] and 77% in another study of children aged 15 months to 5 years [11]. Finally, in another study using electronic monitoring the median adherence among children aged 2–12 years was 84%, but 41% had poor adherence (<80%) [12]. Electronic measurements and pharmacy claims ensure a more accurate adherence estimate [4,10,13] since parents are inclined to overestimate their child’s adherence even when answering anonymously [4,10].

Poor adherence is associated with increased risk of exacerbations and poor disease control [4,6,13,14], which amplifies the importance of improving adherence among children with asthma. This study used a unique access to pharmacy claims during a two-year period to assess adherence to ICS among preschool children with asthma, where adherence studies are scarce.

Adherence in preschool children has mostly been studied in populations including older children, which is problematic as treatment responsibility shifts with increasing age. The aim of this study was to determine possible risk factors for poor adherence in a population of preschool children with asthma, including demographics, atopic comorbidities, exposures, atopic predisposition, and loss of control events.

## 2. Materials and Methods

### 2.1. Study Design and Population

This is a retrospective study of children diagnosed with asthma by a pediatrician at Herlev-Gentofte hospital in Copenhagen, Denmark, from 1 January 2018 to 31 December 2018 (Figure 1). Children diagnosed at either the pediatric emergency department (ED), the inpatient ward or the asthma outpatient clinic were included in the study if they met the following criteria: <6 years of age at diagnosis, asthma diagnosis code (ICD-10), Danish residency, and treatment with ICS at the end of the follow-up period four years from diagnosis [15].

### 2.2. Adherence Assessment

Adherence was assessed in 2022 using the default electronic two-year window available for pharmacy claims in the Danish National Registry. Proportion of days covered (PDC) was calculated for each child treated with ICS as the total number of ICS doses redeemed from the pharmacy divided by the total number of doses needed according to the prescriptions in the two-year period. Changes in prescriptions, discontinuations and dosage adjustments were taken into consideration in the calculations. Poor adherence to ICS was defined as PDC < 80% [7].

### 2.3. Risk Factors and Loss of Control Events

The following baseline characteristics were extracted from the medical records: demographics (sex, age at inclusion, ethnicity, weight, height), birth characteristics (gestational age, birth weight, caesarean section), atopic comorbidity during follow-up (allergic rhinitis, eczema and/or food allergy), atopic predisposition (asthma, allergy and/or eczema in a first degree relative), exposures (passive smoking, furred pets), asthma treatment at inclusion and exacerbations prior to inclusion.

Loss of control events were extracted from the medical records defined as worsening of asthma resulting in a visit to the ED, admission to the pediatric ward, oral corticosteroid (OCS) treatment, step-up in treatment regime according to GINA guidelines [2], short-acting b2-agonist (SABA) redemption or any of the mentioned during the follow-up period. The loss of control events was analyzed dichotomously both yearly and as a total during the four years.

### 2.4. Statistical Analyses

Continuous variables are presented as mean with standard deviation (SD) or median with interquartile range (IQR), and categorical variables are presented as proportions. Comparisons of baseline characteristics were done using Welch’s *t*-test, Wilcoxon Rank Sum test, Chi-squared test, or Fisher’s exact test.

For risk factor analyses, univariate logistic regression was first performed to identify variables independently associated with non-adherence, using a dichotomous PDC outcome (non-adherence: PDC < 80%). Thereafter multivariate logistic regression was applied adjusting for sex, age, and exacerbations prior to inclusion.

To evade a negative skewness of the continuous PDC outcome, a quasi-binomial distribution model was used [7]. This required transformation of PDC to an adherence ratio (AR), which was calculated as days with medical supply divided by days without medical supply. An AR above 1 is interpreted as more days with medical supply than without, i.e., higher adherence, and an AR less than 1 is interpreted as more days without medical supply, i.e., a poorer adherence. A *p*-value < 0.05 was considered statistically significant. All statistical analyses were performed using R free software version 4.1.2.

## 3. Results

### 3.1. Adherence to ICS

A total of 464 children received an asthma diagnosis in 2018, of whom 79 were included in the final analyses (Figure 2). We excluded 245 due to age >6 years, 38 were duplicates, 5 had loss of follow-up data, and 4 had a wrong diagnosis. Further, 93 children were excluded as they did not receive treatment with ICS at end of follow-up or because treatment adherence could not be assessed due to lack of information.

In Table 1, the study population consisting of 79 children were compared to the 93 children, who did not receive treatment with ICS or whose adherence could not be assessed. The study population consisted of 47 males (59.5%) and 52 (65.8%) were of Caucasian ethnicity. The median (IQR) age in the study population was 2.3 years (1.3–3.6), which was significantly higher compared to the excluded children (2.3 years (1.3–3.6) vs. 1.6 years (1.0–2.4), *p* = 0.002). Consequently, the included children’s mean weight was higher than the excluded (13.9 kg (4.4) vs. 12.5 kg (3.3), *p* = 0.031). A total of 58 (84.1%) children had siblings in the study population, which was a significantly larger proportion as compared to the excluded (84.1% vs. 67.7%, *p* = 0.044).

At inclusion, 45 (56.96%) children in the study population received treatment with ICS, which was a significantly higher proportion compared to the excluded (56.96% vs. 30.1%, *p* = 00.001). This was also true for treatment with SABA (98.7% vs. 91.4%, *p* = 0.040) and leukotriene receptor antagonists (LTRA) (16.5% vs. 2.2%, *p* = 00.002) at inclusion. More children in the study population had at least one exacerbation prior to inclusion (69.6% vs. 53.8%, *p* = 00.049) and experienced any type of loss of control event during the follow-up (97.5% vs. 53.8%, *p* < 0.001). This was also true for the different types of events (*p*-values < 0.001).

The median (IQR) adherence in the two-year study period was 41.7% (23.3–79.0) and 59 (74.7%) were classified as non-adherent using of PDC cut-off at 80%.

### 3.2. Risk Factors for Non-Adherence

Adherence to ICS defined as PDC > 80% was associated with having had a loss of control event during the four years of follow-up requiring either admission to a pediatric ward (55.0% vs. 32.2%, _a_OR:0.31 [0.10;0.94], *p* = 00.039) or treatment with OCS (40.0% vs. 15.3%, _a_OR:0.14 [0.04;0.59], *p* = 00.007) in analyses adjusted for sex, age, and exacerbations prior to inclusion. Having had a step-up in preventive treatment (95.0% vs. 74.6%, _a_OR:0.13 [0.02;1.08], *p* = 00.059) and having eczema (64.7% vs. 41.8%, _a_OR:0.34 [0.10;1.16], *p* = 00.084) also showed a trend of association with adherence. The remaining risk factors showed no significant association with adherence in the logistic regression models (Table 2).

Using the continuous AR in models adjusted for sex, age and exacerbations prior to inclusion, showed that a step-up in preventive treatment was significantly associated with increasing adherence (_a_AR:2.34 [1.10;4.98], *p* = 0.03). Increasing adherence was also associated with OCS treatment (_a_AR:2.45 [1.13;5.34], *p* = 0.026), and redeeming a SABA-prescription (_a_AR:2.91 [1.26;6.74], *p* = 0.015). Further, children with an atopic comorbidity had 18% higher adherence compared to children without (_a_AR:1.18 [1.01:1.37], *p* = 0.039). Finally, adherence was 56% lower in children with vs. without a first degree relative with asthma (_a_AR:0.44 [0.23;0.84], *p* = 0.015). None of the other risk factors showed significant association with adherence (Table 3).

Having had any type of loss of control event in the fourth year of follow-up was associated with increasing adherence (_a_AR:2.89 [1.59;5.23], *p* = 0.001), which was also observed for redeeming a SABA prescription (_a_AR:2.93 [1.60;5.36], *p* = 0.001). Having been treated with OCS already in the first year after diagnosis was associated with increasing adherence at the end of follow-up (_a_AR:4.46 [1.38;14.38], *p* = 0.014). These findings were also observed using logistic regression of adherence defined as PDC > 80% (Table 4).

### 3.3. Step-Up in Preventive Treatment and Asthma Control

A total of 63 children had a step-up in treatment during the study period. Of those, 26 (63.4%) were considered well-controlled by a pediatrician after their change in treatment, 15 (36.6%) were not considered well-controlled, and for 22 there was no data on asthma control, e.g., the child had not been seen in the outpatient clinic since the change in treatment.

### 3.4. Lung Function at End of Follow-Up

Lung function was assessed at the end of the follow-up period with different objective tests based on the child’s cooperation in a small proportion of the study population due to their young age (Table 5). Children with non-adherence (PDC < 80%) had a higher mean (SD) specific airway resistance, sR_aw_: 1.43 kPa/s (0.64), compared to adherent children, sR_aw_: 1.06 (0.15), i.e., worse lung function, although this was not statistically significant (*p* = 0.114).

## 4. Discussion

### 4.1. Main Findings

This study found that almost three quarter of preschool children with asthma were non-adherent to ICS treatment. Children, who had experienced a loss of control event requiring either OCS, a step up in preventive treatment according to GINA guidelines or redeeming a SABA prescription, were more adherent to their treatment with ICS. Further, children with an atopic comorbidity were more adherent to their ICS treatment, whereas children with a first degree relative with asthma, were less adherent.

### 4.2. Detailed Findings

#### 4.2.1. Adherence Assessment

The median adherence measured as PDC in our study was only 41.7%, which may be due to some parents reducing or eliminating ICS during asymptomatic periods or parents who prefer non-pharmacological strategies including environmental control. In studies by Burgess et al. and Gibson et al. the median adherence among children aged 15 months to 7 years was 70.5% and 77%, respectively [10,11]. These percentages were calculated by dividing the number of inhaled doses registered using an electronic monitoring device attached to the inhaler with the total amount prescribed, contrasting our study where adherence was calculated as the total amount of ICS doses redeemed from the pharmacy divided by the total amount prescribed. Klok et al. and Lasmar et al. included children from 1–12 years and found a median adherence of 84% and 63%, the latter using pharmacy records to assess adherence as the number of doses refilled at a specific pharmacy divided by the total prescribed for two years [12,16]. In an even wider age group (0–18 years), Sherman et al. and Elkout et al., estimated adherence to ICS to 44% and 51% using pharmacy records of refilled prescriptions [17,18]. Older children have poorer adherence to treatment [5,7,8], and the adherence estimate in the studies including older children could therefore underestimate the adherence in the younger children included in the studies. However, as the findings above indicate, there is a large span of adherence estimates across studies, which makes generalization of adherence difficult. The use of different assessment methods, either using the number of actual inhalations monitored by a device attached to the inhaler or using pharmacy records of prescriptions indicating the number of days with medical supply, additionally makes comparison of the findings difficult.

In our study, the included children were older than the children, who were excluded due to not being treated with ICS. The included children also weighed more and more had siblings, which is likely due to the children being older. More children had a loss of control event, more received treatment at inclusion, and more had had an exacerbation prior to inclusion. The study population therefore consists of children with more severe asthma, who require preventive treatment. However, this is the study population we are most interested in, since interventions in this patient group, particularly, have the possibility to reduce asthma morbidity.

#### 4.2.2. Loss of Control

Poor adherence is associated with poor disease control and risk of exacerbations among school-aged children and adolescents [4,6,13,14]. In our study, we found that the loss of control event itself is associated with better adherence to treatment with ICS and that the more recent a loss of control event is, the more children are adherent. In line with our findings, a study by Williams et al. among patients 12–56 years of age found that adherence to ICS increased following an exacerbation of either an asthma-related hospitalization, an ED visit or treatment with OCS [19]. Further, a focus group interview among adolescents aged 12–20 years indicated an improvement in adherence to their treatment following an exacerbation, which made them aware of the severity of their asthma [20]. In contrast, Lasmar et al. investigated children aged 1–12 years and found no difference in adherence following an exacerbation [16]. However, Lasmar et al. used a cut-off value for adherence of 70%, whereas ours was 80%.

A step-up in preventive treatment according to GINA guidelines was positively associated with adherence in our study. The step-up was mainly an increase in the daily ICS dosage from low to medium dose, and not add on of other medications, e.g., LTRA. In contrast, studies among children aged 1–17 years with moderate to severe asthma requiring daily ICS treatment have shown that twice daily dosing or an increase of puffs daily was associated with poorer adherence to ICS treatment [16,21]. Further, a randomized trial by Mallol et al. also found an association between twice daily dosing and poorer adherence compared to once daily dosing among children with mild to moderate persistent asthma aged 7–16 years [22]. However, in the study by Mallol et al. the total dosage was the same in the two groups, who only differed in the frequency of administering ICS. The poor adherence found in these studies is therefore more likely attributed to the inconvenience of twice daily dosing rather than the increase of the dosage itself. In our study, 63.4% of the children, who had a step-up, gained better control of their asthma, and were classified as well-controlled after the change in treatment, which illustrates the effect of better adherence as well as having a step-up in treatment.

#### 4.2.3. Atopic Comorbidity

We found that children with any atopic comorbidity, including allergic rhinitis, eczema, food allergy and allergic sensitization to pets and house-dust mites were more adherent to their ICS treatment compared to children without. Similarly, Lasmar et al. showed that absence of allergic rhinitis among children with asthma was associated with poorer adherence [16].

#### 4.2.4. Predisposition to Asthma

Surprisingly, we observed that children with a first degree relative with asthma had poorer adherence compared to children without. Adherence is generally associated with being knowledgeable about asthma and the effect of treatment [3,4,9,23], which would be expected especially among parents, who have asthma themselves. Koster et al. also found a trend similar to our study towards poorer adherence if a parent had asthma [8]. Further, having a family member with asthma has been found associated with poorer asthma control, where a possible explanation was assumed to be competing priorities in the household interfering with the parent’s knowledge of their child’s asthma symptoms and treatment [24].

In a study of migraine in a pediatric population, Eidlitz-Markus et al. found that children of parents with migraine had a significantly longer duration of symptoms before admission to a headache clinic [25]. In this case, the assumption was that the parents are more familiar with the symptoms and therefore less anxious. These results together with our findings could indicate that parents with the same chronic illness have a different disease severity threshold for reacting, which could influence their behaviors regarding adherence, treatment, and contact with the healthcare system.

The findings in our study could also be spurious, since having a first degree relative with eczema or allergy showed an insignificant association with adherence.

#### 4.2.5. Lung Function

The children classified as non-adherent (PDC < 80%) had an estimated specific airway resistance of 1.43 kPa/s, which was higher than that for the adherent children, who had an estimate of 1.06 kPa/s. This finding was not statistically significant but is an indication of the effect of adherence to ICS on lung function.

### 4.3. Strengths and Limitations

A significant strength of the study is that we were able to collect all prescriptions filled from any Danish pharmacy without restrictions to use of a specific pharmacy, which is typically a limitation when using data from pharmacy claims [4]. Adherence was assessed for the index child included. However, if the child had other family members with asthma as well, and the parents only filled other family members prescriptions at the pharmacy, the adherence would be underestimated. Further, the family could also share the child’s inhalers in which case the adherence would be overestimated.

Another limitation using pharmacy claims is that we cannot determine if the child is taking the prescribed medication. However, the 2-year observation period limits this risk since the prescription is collected multiple times and thus strengthens the probability that the child uses the medication. Electronic measurements of using the inhaler would provide more certainty that the medication is taken, but this could also influence the adherence, since the parents and child are aware that their adherence is being monitored [10,11]. Using pharmacy claims in our study therefore limits the risk of influencing adherence, and the results may therefore be more reflective of the everyday setting.

The current study population consisted of 79 preschool-aged children. The subgroups in the risk factor analyses, where the population is divided according to their adherence, are therefore small. However, analyzing the findings both as a dichotomous variable using a PDC cut-off at 80% and as a continuous variable using AR strengthens the results with similar findings, which were stronger using the continuous outcome.

We were not able to assess socioeconomic status in this study such as parents’ income, employment status, and level of education. The latter has in some studies been associated with adherence [16,23]. However, the effect of socioeconomic status on adherence, and whether this is associated with poorer adherence or not, is still not fully determined [3].

## 5. Conclusions

Approximately three quarter of children aged <6 years in this study were non-adherent to ICS. Children, who redeemed a SABA prescription, had a step-up in preventive treatment, or were treated with OCS in the study period, were more adherent. Further, children with atopic comorbidity were more adherent, whereas children with a first degree relative with asthma had poorer adherence. These findings could be valuable in clinical practice for identifying children at risk of non-adherence to preventive asthma treatment.

## Figures and Tables

**Figure 1 children-10-00043-f001:**
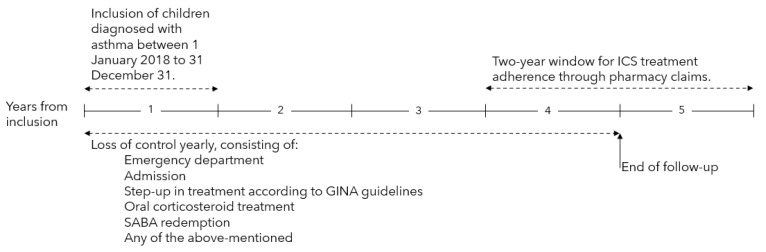
Timeline of the study.

**Figure 2 children-10-00043-f002:**
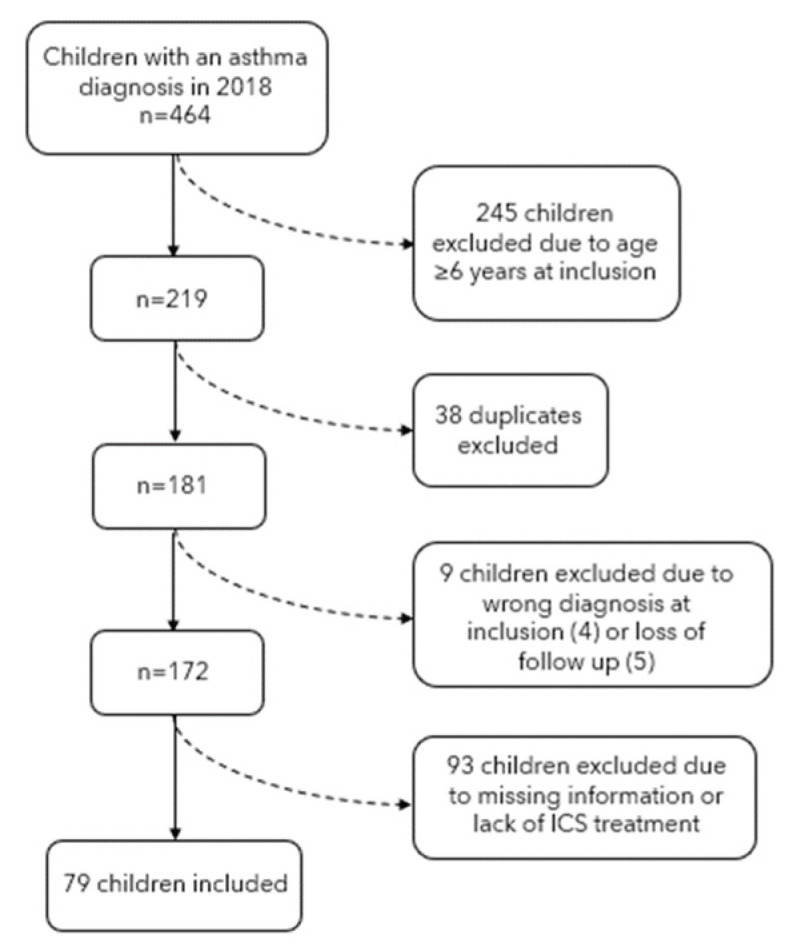
Flowchart of children included in the study.

**Table 1 children-10-00043-t001:** Comparison of the included (n = 79) and excluded (n = 93).

	Included (n = 79)	Missing Data	Excluded (n = 93)	Missing Data	*p* Value
Sex (Male), n (%)	47 (59.5%)	0	58 (62.4%)	0	0.820
Age (years), median (IQR)	2.3 (1.3–3.6)	0	1.6 (1.0–2.4)	0	0.002 *
Siblings (Yes), n (%)	58 (84.1%)	10	44 (67.7%)	28	0.044
Caucasian ethnicity (Yes), n (%)	52 (65.8%)	0	59 (63.4%)	0	0.869
Weight (kg), mean (SD)	13.9 (4.4)	3	12.5 (3.3)	9	0.031
Height (cm), mean (SD)	94.2 (22.3)	41	85.7 (14.7)	58	0.058
Gestation Gestational age (days), median (IQR) Birthweight (g), mean (SD) C-section (Yes), n (%)	273 (261.5–281) 3197 (790.1) 24 (35.8%)	8 9 12	277 (266.8–285.5) 3315.4 (795.6) 19 (26.4%)	13 12 21	0.180 * 0.362 0.308
Exacerbations prior to inclusion (Yes), n (%)	55 (69.6%)	0	50 (53.8%)	0	0.049
Exposures (Yes), n (%) Tobacco Cat/Dog	3 (6.5%) 6 (17.1%)	33 44	1 (3.1%) 3 (12.0%)	61 68	0.640 ** 0.722 **
Comorbidities (Yes), n (%) Any ^1^ Allergic rhinitis Eczema Food allergy	44 (61.1%) 33 (55.0%) 34 (47.2%) 10 (17.2%)	7 19 7 21	33 (51.6%) 11 (34.4%) 26 (44.8%) 6 (14.3%)	29 61 35 51	0.343 0.096 0.924 0.903
Treatment at inclusion (Yes), n (%) ICS SABA LTRA	45 (56.96%) 78 (98.7%) 13 (16.5%)	0 0 0	28 (30.1%) 85 (91.4%) 2 (2.2%)	0 0 0	0.001 0.040 ** 0.002
Predisposition ^2^ (Yes), n (%) Any Asthma Allergy Eczema	57 (79.2%) 39 (54.2%) 34 (47.9%) 15 (21.1%)	7 7 8 8	41 (67.2%) 28 (47.5%) 25 (43.9%) 15 (26.8%)	32 34 36 37	0.173 0.556 0.783 0.593
Loss of control (Yes), n (%) Any ED Admission Step-up Oral corticosteroid SABA redemption	77 (97.5%) 49 (62.0%) 30 (37.97%) 63 (79.7%) 17 (21.5%) 66 (83.5%)	0 0 0 0 0 0	50 (53.8%) 30 (32.3%) 8 (8.6%) 24 (25.8%) 2 (2.2%) 23 (24.7%)	0 0 0 0 0 0	<0.001 <0.001 <0.001 <0.001 <0.001 <0.001

^1^ Allergic rhinitis, eczema, food allergy, pollen allergy, sensibilization to dust mites and pets, ^2^ First degree relative * Wilcoxon Rank Sum test. ** Fisher’s exact test. ED = Emergency department, SD = Standard deviation, IQR = interquartile range, ICS = Inhaled corticosteroid, SABA = short acting B-agonist, LTRA = Leukotriene antagonist.

**Table 2 children-10-00043-t002:** Risk factors related to non-adherence (PDC < 80%).

	PDC > 80%		PDC < 80%			Crude			Adjusted ^1^		
	n = 20	Missing Data	n = 59	Missing Data	*p* Value	_c_OR	[95%CI]	*p* Value	_a_OR	[95%CI]	*p* Value
Sex (Male), n (%)	10 (50.0%)	0	37 (62.7%)	0	0.461	1.68	[0.60;4.68]	0.319	1.70	[0.60;4.78]	0.317
Age (years), median (IQR)	2.2 (1.2–4.0)	0	2.4 (1.3–3.4)	0	0.933 *	0.99	[0.73;1.36]	0.970	1.02	[0.73;1.43]	0.906
Siblings (Yes), n (%)	14 (87.5%)	4	44 (83.0%)	6	1**	0.70	[0.13;3.62]	0.669	0.58	[0.10;3.38]	0.549
Caucasian ethnicity (Yes), n (%)	13 (65.0%)	0	39 (66.1%)	0	1	1.05	[0.36;3.05]	0.929	1.12	[0.38;3.31]	0.838
C-section (Yes), n (%)	7 (36.8%)	1	17 (35.4%)	11	1	0.94	[0.31;2.84]	0.913	0.92	[0.30;2.80]	0.884
Exacerbations prior to inclusion (Yes), n (%)	14 (70.0%)	0	41 (69.5%)	0	1	0.98	[0.32;2.95]	0.966	0.98	[0.32;3.13]	0.970
Exposure, (Yes), n (%) Tobacco Cat/Dog	0 (0.0%) 1 (9.1%)	9 9	3 (8.6%) 5 (20.8%)	24 35	-- 0.640 **	-- 2.63	-- [0.27;25.72]	-- 0.405	-- 2.53	-- [0.18;36.06]	-- 0.494
Comorbidities (Yes), n (%) Any ^2^ Allergic rhinitis Eczema Food allergy	10 (55.6%) 8 (53.3%) 11 (64.7%) 1 (7.1%)	2 5 3 6	34 (63.0%) 25 (55.6%) 23 (41.8%) 9 (20.5%)	5 14 4 15	0.780 1 0.169 0.424 **	1.36 1.09 0.39 3.34	[0.46;4.01] [0.34;3.53] [0.13;1.21] [0.38;29.03]	0.577 0.881 0.104 0.274	1.33 0.87 0.34 3.10	[0.40;4.35] [0.24;3.17] [0.10;1.16] [0.33;28.87]	0.642 0.831 0.084 0.320
Treatment at inclusion (Yes), n (%) ICS SABA LTRA	12 (60.0%) 19 (95.0%) 4 (20.0%)	0 0 0	33 (55.9%) 59 (100%) 9 (15.3%)	0 0 0	0.955 0.253 ** 0.729 **	0.85 -- 0.72	[0.30;2.37] -- [0.20;2.66]	0.751 -- 0.622	0.83 -- 0.80	[0.27;2.63] -- [0.20;3.17]	0.758 -- 0.755
Predisposition ^3^ (Yes), n (%) Any Asthma Allergy Eczema	15 (75.0%) 8 (40.0%) 11 (55.0%) 5 (25.0%)	0 0 0 0	42 (80.8%) 31 (59.6%) 23 (45.1%) 10 (19.6%)	7 7 8 8	0.747 ** 0.218 0.626 0.748 **	1.40 2.21 0.67 0.73	[0.41;4.76] [0.77;6.34] [0.24;1.90] [0.21;2.49]	0.590 0.139 0.454 0.617	1.38 2.11 0.54 0.77	[0.38;4.98] [0.71;6.23] [0.18;1.66] [0.21;2.83]	0.626 0.176 0.283 0.694
Loss of control (Yes), n (%) Any ED Admission Step-up Oral corticosteroid SABA redemption	20 (100%) 14 (70.0%) 11 (55.0%) 19 (95.0%) 8 (40.0%) 18 (90.0%)	0 0 0 0 0 0	57 (96.6%) 35 (59.3%) 19 (32.2%) 44 (74.6%) 9 (15.3%) 48 (81.4%)	0 0 0 0 0 0	-- 0.559 0.121 0.058 ** 0.029 ** 0.498	-- 0.63 0.39 0.15 0.27 0.48	-- [0.21;1.86] [0.14:1.10] [0.02;1.25] [0.09;0.85] [0.10;2.40]	-- 0.397 0.074 0.080 0.025 0.376	-- 0.56 0.31 0.13 0.14 0.46	-- [0.18;1.73] [0.10;0.94] [0.02;1.08] [0.04;0.59] [0.09;2.31]	-- 0.314 0.039 0.059 0.007 0.345

^1^ Sex, age, exacerbations prior to inclusion. ^2^ Allergic rhinitis, eczema, food allergy, pollen allergy, sensibilization to dust mites and pets. ^3^ First degree relative. * Wilcoxon Rank Sum test. ** Fisher’s exact test. ED = emergency department, IQR = interquartile range, ICS = inhaled corticosteroids, SABA = short acting B-agonist, LTRA = leukotriene antagonist.

**Table 3 children-10-00043-t003:** Adherence ratio (AR), days with ICS treatment/days without ICS treatment.

	Crude			Adjusted ^1^			
	_c_AR	95%CI	*p* Value	_a_AR	95%CI	*p* Value	Missing Data
Sex (Male)	0.90	[0.50;1.61]	0.716	0.91	[0.50;1.66]	0.762	0
Age (years)	1.04	[0.87;1.25]	0.636	1.00	[0.83;1.22]	0.975	0
Siblings (Yes)	1.31	[0.56;3.04]	0.535	1.27	[0.51;3.20]	0.610	10
Caucasian ethnicity (Yes)	0.96	[0.52;1.76]	0.898	0.98	[0.52;1.83]	0.943	0
C-section (Yes)	1.06	[0.55;2.04]	0.859	1.03	[0.53;2.01]	0.933	12
Exacerbations prior to inclusion (Yes)	1.51	[0.81;2.83]	0.201	1.50	[0.77;2.93]	0.238	0
Exposure Tobacco Cat/Dog	0.95 0.39	[0.22;4.12] [0.11;1.31]	0.948 0.136	2.26 0.28	[0.44;11.67] [0.07;1.21]	0.334 0.099	33 44
Comorbidities (Yes) Any ^2^ Allergic rhinitis Eczema Food allergy	1.11 1.09 1.69 0.69	[0.60;2.05] [0.58;2.07] [0.93;3.06] [0.29;1.64]	0.745 0.789 0.088 0.406	1.18 1.30 1.71 0.80	[1.01;1.37] [0.63;2.69] [0.90;3.23] [0.32;1.99]	0.039 0.479 0.104 0.638	7 19 7 21
Treatment at inclusion (Yes) ICS SABA LTRA	1.16 0.10 0.93	[0.65;2.08] [0.001;7.86] [0.43;2.02]	0.610 0.309 0.855	0.98 0.07 0.80	[0.51;1.90] [0.001;5.82] [0.35;1.82]	0.962 0.240 0.592	0 0 0
Predisposition ^3^ (Yes) Any Asthma Allergy Eczema	0.51 0.45 1.22 1.69	[0.24;1.10] [0.25;0.83] [0.66;2.27] [0.78;3.66]	0.092 0.013 0.529 0.184	0.47 0.44 1.22 1.65	[0.21;1.07] [0.23;0.84] [0.63;2.35] [0.72;3.76]	0.078 0.015 0.556 0.237	7 7 8 8
Loss of control (Yes) Any ED Admission Step-up Oral corticosteroid SABA redemption	1.53 1.18 1.61 2.16 2.27 2.73	[0.24;9.94] [0.65;2.13] [0.89;2.92] [1.04;4.47] [1.10;4.65] [1.20;6.18]	0.657 0.593 0.118 0.043 0.029 0.019	1.77 1.24 1.71 2.34 2.45 2.91	[0.26;11.90] [0.67;2.30] [0.91;3.21] [1.10;4.98] [1.13;5.34] [1.26;6.74]	0.560 0.500 0.097 0.030 0.026 0.015	0 0 0 0 0 0

^1^ Sex, age, exacerbations prior to inclusion. ^2^ Allergic rhinitis, eczema, food allergy, pollen allergy, sensibilization to dust mites and pets. ^3^ First degree relative. ICS = inhaled corticosteroids, SABA = short acting B agonist, LTRA = leukotriene antagonist, ED = emergency department.

**Table 4 children-10-00043-t004:** Yearly loss of control.

	PDC < 80%, n (%)			Non-Adherence ^1^			Adherence Ratio ^1^			Missing Data
	0	≥1	*p* Value	_a_OR	[95%CI]	*p* Value	_a_AR	[95%CI]	*p* Value	
Any (Yes) 1 2 3 4	39 (72.2%) 33 (75.0%) 41 (71.9%) 29 (64.4%)	20 (80.0%) 26 (74.3%) 18 (81.8%) 30 (88.2%)	0.645 1 0.537 0.032	0.58 0.99 0.58 0.23	[0.17;2.00] [0.35;2.81] [0.16;2.05] [0.07;0.78]	0.388 0.978 0.396 0.019	1.31 1.40 1.58 2.89	[0.66;2.59] [0.77;2.56] [0.80;3.12] [1.59;5.23]	0.443 0.278 0.192 0.001	0 0 0 0
ED (Yes) 1 2 3 4	30 (71.4%) 15 (71.4%) 10 (66.7%) 6 (60.0%)	29 (78.4%) 44 (75.9%) 49 (76.6%) 53 (76.8%)	0.653 0.914 0.512 0.263	0.61 0.75 0.56 0.41	[0.20;1.84] [0.24;2.36] [0.16;1.97] [0.10;1.72]	0.379 0.621 0.367 0.223	1.08 1.26 1.11 2.51	[0.58;2.01] [0.64;2.48] [0.52;2.38] [0.98;6.45]	0.815 0.505 0.780 0.059	0 0 0 0
Admission (Yes) 1 2 3 4	12 (63.2%) 6 (60.0%) 1 (100.0%) 5 (62.5%)	47 (78.3%) 53 (76.8%) 58 (74.4%) 54 (76.1%)	0.239 0.263 -- 0.410	0.37 0.37 -- 0.47	[0.11;1.25] [0.09;1.58] -- [0.10;2.25]	0.109 0.178 -- 0.343	1.61 1.17 0.89 2.06	[0.78;3.32] [0.47;2.88] [0.06;13.65] [0.74;5.70]	0.200 0.737 0.935 0.170	0 0 0 0
Step-up (Yes) 1 2 3 4	21 (77.8%) 23 (79.3%) 13 (61.9%) 5 (62.5%)	38 (73.1%) 36 (72.0%) 46 (79.3%) 54 (76.1%)	0.855 0.651 0.201 0.410	1.25 1.43 0.42 0.50	[0.40;3.86] [0.47;4.35] [0.14;1.29] [0.10;2.45]	0.699 0.532 0.129 0.395	0.95 0.99 1.63 2.25	[0.50;1.79] [0.53;1.84] [0.82;3.22] [0.80;6.34]	0.869 0.971 0.164 0.129	0 0 0 0
Oral corticosteroid (Yes) 1 2 3 4	3 (37.5%) 5 (62.5%) 1 (100.0%) 3 (75.0%)	56 (78.9%) 54 (76.1%) 58 (74.4%) 56 (74.7%)	0.022 0.410 -- 1	0.08 0.40 -- 0.92	[0.01;0.47] [0.08;2.09] -- [0.09;9.65]	0.005 0.276 -- 0.944	4.46 1.24 0.89 1.48	[1.38;14.38] [0.44;3.46] [0.06;13.65] [0.38;5.79]	0.014 0.682 0.935 0.579	0 0 0 0
SABA redemption (Yes) 1 2 3 4	-- 14 (77.8%) 36 (73.5%) 24 (63.2%)	-- 25 (71.4%) 23 (76.7%) 35 (85.4%)	-- 0.748 0.960 0.045	-- 1.79 0.84 0.26	-- [0.43;7.48] [0.29;2.45] [0.08;0.83]	-- 0.423 0.748 0.023	-- 1.13 1.79 2.93	-- [0.49;2.64] [0.98;3.29] [1.60;5.36]	-- 0.770 0.064 0.001	79 26 0 0

^1^ Adjusted for sex, age, exacerbations prior to inclusion. ED = Emergency department, SABA = short acting B-agonist.

**Table 5 children-10-00043-t005:** Adherence and lung function at end follow-up.

	PDC > 80% n = 20	Missing Data	PDC < 80% n = 59	Missing Data	*p* Value
FEV_1_ (%) ^1^, mean (SD)	97.17 (11.60)	14	98.11 (13.42)	50	0.887
FEV_1_/FVC ratio, mean (SD)	81.72 (8.28)	14	82.13 (6.59)	50	0.921
sR_aw_ (kPa/s), mean (SD)	1.06 (0.15)	16	1.43 (0.64)	49	0.114

^1^ Percentage of expected FEV_1._

## Data Availability

All data are available on request from the corresponding author.

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
