# Peer review of "Risk Factors for Non-Adherence to Inhaled Corticosteroids in Preschool Children with Asthma"

_children, 2022, doi:10.3390/children10010043_

Round 1

Reviewer 1 Report

The following points need to be addressed in a revised version of the manuscript.

1-    In Introduction Authors should mention that treatment recommendations for children of 5 years of age or younger are based on the available evidence and on expert opinion. Although the evidence is expanding it is still rather limited as most clinical trials in this age group have not characterized participants with respect to their symptom pattern, and different studies have used different outcomes and different definitions of exacerbations.

2-    Authors should also mention how much the choice of inhaler device is also an important consideration, which may impact on adherence.

3-    Many adolescents and parents adapt inhaled corticosteroids use according to the prevalence of asthma symptoms, by reducing or eliminating controller medication in the absence of symptoms.

4-    A considerable percentage of parents expressed fear of side effects of inhaled corticosteroids, although the impact of these concerns on adherence is unclear. 

5-    Some parents may prefer non-pharmacological strategies including environmental control where appropriate

Author Response

The following points need to be addressed in a revised version of the manuscript.

  1. In Introduction Authors should mention that treatment recommendations for children of 5 years of age or younger are based on the available evidence and on expert opinion. Although the evidence is expanding it is still rather limited as most clinical trials in this age group have not characterized participants with respect to their symptom pattern, and different studies have used different outcomes and different definitions of exacerbations.

RESPONSE: We have now added the following:

Page 1, lines 40-43: Preventive treatment with inhaled corticosteroids (ICS) is the cornerstone treatment of asthma, which is administered daily and required also when the child is asymptomatic. ICS is also the primary asthma controller of choice for children of 5 years and younger although the evidence base is still rather limited and treatment decisions in some cases rely on expert opinions(2).

  1. Authors should also mention how much the choice of inhaler device is also an important consideration, which may impact on adherence.

RESPONSE: Thanks for this suggestion. We have added the following:

Page 1, lines 43-45: Maintaining adherence to ICS may depend on choice of inhaler device and is generally challenging as studies show poor adherence in approximately 50% of children with asthma aged 8-16 years(3–6).

  1. Many adolescents and parents adapt inhaled corticosteroids use according to the prevalence of asthma symptoms, by reducing or eliminating controller medication in the absence of symptoms.

RESPONSE: Good point. The text has been modified as follows:

Page 2, lines 48-51: Among preschool children, the parents are responsible for administering treatment(9) and may by prone to reduce or eliminate controller medication in the absence of symptoms, particularly due to fear of side effects of ICS.

  1. A considerable percentage of parents expressed fear of side effects of inhaled corticosteroids, although the impact of these concerns on adherence is unclear.

RESPONSE: We have now included this important consideration:

Page 2, lines 48-51: Among preschool children, the parents are responsible for administering treatment(9) and may by prone to reduce or eliminate controller medication in the absence of symptoms, particularly due to fear of side effects of ICS.

  1. Some parents may prefer non-pharmacological strategies including environmental control where appropriate.

RESPONSE: The following has been added to the discussion section:

Page 9, lines 216-18: “The median adherence measured as PDC in our study was only 41.7%, which may be due to some parents reducing or eliminating ICS during asymptomatic periods or parents who prefer non-pharmacological strategies including environmental control.

Reviewer 2 Report

To the Authors

Summary

This study investigated risk factors associated with non-adherence to asthma therapy in a sample of 79 preschool children with asthma. Results indicated that approximately 75% of children were non-adherent and that adherence was positively associated with loss of control requiring step-up treatment with an asthma controller or oral ICS and having an atopic co-morbidity. Contrastingly, having a first-degree relative with asthma was associated with worse adherence. The authors concluded that consideration of these factors could improve medication adherence in pre-schoolers with asthma.

Overall, this manuscript was well-written and comprehensible for a wide range of readers, including medical and non-medical researchers. The introduction outlined the rationale for the study. The study design, methodology, and statistical tests were appropriate. Data was presented clearly in tables. Potential explanations for the findings were amply argued and supported by evidence from the relevant literature. In addition, study limitations were provided.

The authors are to be commended and we look forward to reading more asthma studies from this research group focusing on novel approaches for optimizing adherence to asthma therapy in young patients

Comment: Only one minor point, the citation of references in the text does not conform to the author's guidelines [ ].

Author Response

This study investigated risk factors associated with non-adherence to asthma therapy in a sample of 79 preschool children with asthma. Results indicated that approximately 75% of children were non-adherent and that adherence was positively associated with loss of control requiring step-up treatment with an asthma controller or oral ICS and having an atopic co-morbidity. Contrastingly, having a first-degree relative with asthma was associated with worse adherence. The authors concluded that consideration of these factors could improve medication adherence in pre-schoolers with asthma.

Overall, this manuscript was well-written and comprehensible for a wide range of readers, including medical and non-medical researchers. The introduction outlined the rationale for the study. The study design, methodology, and statistical tests were appropriate. Data was presented clearly in tables. Potential explanations for the findings were amply argued and supported by evidence from the relevant literature. In addition, study limitations were provided.

The authors are to be commended and we look forward to reading more asthma studies from this research group focusing on novel approaches for optimizing adherence to asthma therapy in young patients.

RESPONSE: Thanks for the positive feedback.

Comment: Only one minor point, the citation of references in the text does not conform to the author's guidelines [ ].

RESPONSE: This has been corrected throughout the text.